# Effects of woody plant encroachment on abundance of multiple tick species in the U.S. Great Plains

Jozlyn Propst[1]*, Tucker Taylor[1], Scott R. Loss[2], Bruce H. Noden[1]

**1** Department of Entomology and Plant Pathology, Oklahoma State University, Stillwater, Oklahoma, United States of America, **2** Department of Natural Resource Ecology and Management, Oklahoma State University, Stillwater, Oklahoma, United States of America

* jozlyn.d.kizer@okstate.edu

## Abstract

Vector-borne diseases are increasing in prevalence, posing a risk to 80% of the human population worldwide. In the United States, tick-borne diseases account for 75% of all nationally reported vector-borne disease cases. As the distribution of medically important ticks and tick-borne diseases increases, it is crucial to understand the factors driving their expansion. Changes in land cover are a known driver of vector-borne disease emergence and prevalence. In the U.S. Great Plains, a dominant form of land cover change affecting vectors and vector-borne diseases is woody plant encroachment (WPE), the expansion of trees and shrubs into grassland ecosystems due to fire suppression and other anthropogenic factors. In this study, we examined whether WPE by eastern redcedar (*Juniperus virginiana*, ERC), a widespread encroaching tree in the Great Plains, influences abundance of three medically important tick species, *Amblyomma americanum*, *Dermacentor variabilis*, and *Amblyomma maculatum*. Through $CO_2$ trapping and flagging, we collected ticks in Oklahoma, USA, in sites capturing open grasslands and three intensifying stages of ERC encroachment. For all analyses, we found that abundances of *A. americanum* and *D. variabilis* were higher in the earliest stage of ERC encroachment compared to open grassland, indicating that abundances of these species increase shortly after the onset of encroachment. However, for *A. maculatum*, a species known to be associated with grassland ecosystems, abundance was lower in areas experiencing ERC encroachment compared to grasslands. Our results suggest that the most effective strategy to managing tick populations in association with WPE is to prevent ERC encroachment entirely or to remove encroachment early on before progression to later stages. Landowners, managers of public and private lands, and public health officials should be aware of the health risks associated with ERC encroachment and consider initiating educational and management protocols to protect their communities.

**Data availability statement:** All relevant data are within the manuscript and its supporting information files.

**Funding:** Funding for the research was provided by the National Institutes of Health (R03-5R03AI163283-02 [BHN and SRL]) (https://www.nih.gov/) and the Tick Rearing Facility (OKL-0336 [BHN]), the OSU 2024 President's Fellows Faculty Research Award and the U.S. Department of Agriculture National Institute of Food and Agriculture through the Oklahoma Agricultural Experiment Station from a Hatch (OKL-03085 [BHN] and OKL-03150 [SRL]) (https://nifa.usda.gov/) and Multistate (NE-2443 [BHN]) projects. Funders did not play any role in the study design, data collection and analysis, decision to publish, or preparation of the manuscript.

**Competing interests:** The authors have declared that no competing interests exist.

## Introduction

Globally, vector-borne diseases are increasing in prevalence, posing a risk to 80% of the human population and contributing to approximately 700,000 annual deaths worldwide [1]. Tick-borne diseases are the most prevalent vector-borne diseases in the United States, making up >75% of nationally reported cases between 2004 and 2016 [2]. Over the past two decades, the United States has seen an increase in several tick-borne diseases including ehrlichiosis and spotted fever group rickettsiosis, as well as two newly emerging tick-borne viruses, Bourbon virus and Heartland virus [2–4]. Anthropogenic land cover change is a major contributing factor in the ecology of vector-borne disease systems and the transmission of disease to humans [5]. Further research into the effects of land cover change on all aspects of these disease systems, including vectors, hosts, pathogens, and their surrounding abiotic conditions, is necessary to enhance understanding of the emergence and rising prevalence of tick-borne diseases in the U.S.

In grassland ecosystems around the world, a major type of human-caused land cover change that appears to be influencing vector-borne disease transmission is woody plant encroachment (WPE). WPE is the proliferation of native or non-native woody plant species into grassland and shrubland ecosystems, and is caused by factors such as fire suppression, intensive grazing, drought, and climate change [6–8]. WPE changes all aspects of the grassland ecosystems it affects, including abiotic conditions, vegetation cover, and animal population abundance and community composition [9,10]. Expansion of woody plants in grasslands reduces abundance and diversity of grassland-dependent species, increases abundance of habitat-generalists and can result in monocultural closed-canopy landscapes that are relatively low in species diversity [6,9,11–15]. Although further research is needed to identify abiotic effects of WPE, this process can increase air humidity and affect soil moisture through its changes to rates of evapotranspiration and precipitation interception, infiltration, and runoff, all of which results in substantial abiotic changes in encroached areas compared to open grasslands [16–18]. All these changes brought about by WPE have the potential to influence populations of pathogen vectors, hosts, vector-host interactions, and thus the overall nidus of disease transmission on the landscape [19].

In the Great Plains of the United States, eastern redcedar (*Juniperus virginiana,* ERC) (Fig 1) is one of the principal encroaching woody species, and is causing wholesale, landscape-level transformations of grasslands into extensive monocultural woodlands and forests across a broad swath from Texas to the Dakotas [20]. ERC is readily top-killed by fire; however, widespread fire suppression across the Great Plains has allowed this native evergreen, characterized by rapid growth and seed dispersal facilitated by birds and small mammals, to expand into grasslands, converting them into closed-canopy forests in as little as 20–30 years [6,7,21]. Additionally, ERC modifies abiotic conditions by increasing relative humidity and decreasing wind speed compared to surrounding open grasslands [20]. ERC also changes species composition of wildlife communities, including those of birds and mammals, resulting in a shift from dominance by grassland-dependent species to generalist and woodland-dependent species, some of which are documented tick

## Eastern Redcedar Encroachment Stages

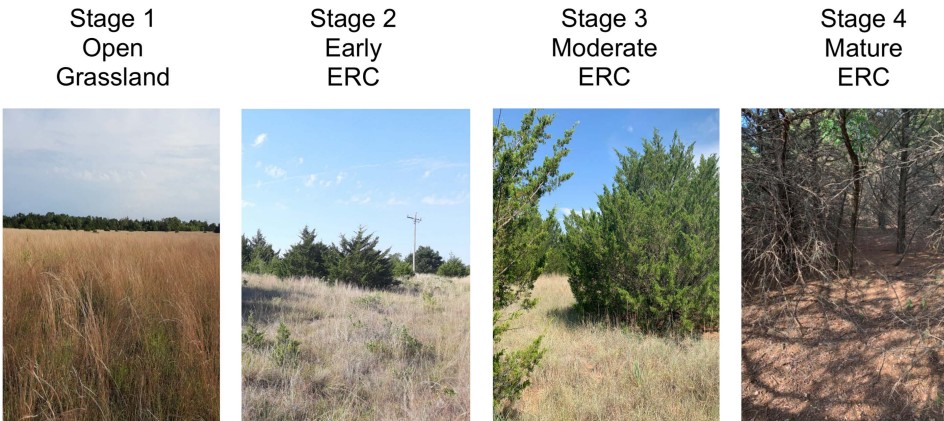

| Stage 1 Open Grassland | Stage 2 Early ERC | Stage 3 Moderate ERC | Stage 4 Mature ERC |

**Fig 1. Examples of increasing levels of eastern redcedar encroachment into grasslands in Oklahoma, USA.**

bloodmeal hosts and/or pathogen reservoir hosts (e.g., white-tailed deer [*Odocoileus virginianus*], white-footed mouse [*Peromyscus leucopus*], and several bird species) [14,22]. Notably, the presence of ERC has also been shown to increase the abundance of ticks, and may increase prevalence of tick-borne pathogens, compared to un-encroached grasslands [23–25]. However, past research into ERC effects on ticks and pathogens has been characterized by a relatively limited spatial and temporal scope with comparisons of tick abundance under individual ERC trees to abundance in nearby open areas, and sampling covering a single season. Longer-term research is needed to confirm the effects of ERC on ticks and tick-borne disease systems, and to evaluate how tick abundance changes at broader spatial scales in different stages of encroachment.

We addressed this research gap regarding the impacts of WPE by ERC on the ecology of tick-borne diseases by evaluating how tick abundance varies among sites capturing 4 different stages of ERC encroachment in Oklahoma, USA. Oklahoma is characterized by a high prevalence of several tick-borne diseases [26,27] and is also experiencing some of the greatest rates of ERC encroachment [28], making it an ideal study area for addressing this goal. Our specific objective was to analyze effects of ERC encroachment stage on numbers of three different tick species (*Amblyomma americanum*, *Dermacentor variabilis*, and *A. maculatum*). Based on previous research suggesting increased abundance of *A. americanum* and *D. variabilis* under ERC trees [23,24], we hypothesized that abundance of these two species would increase with each successive stage of ERC encroachment, from open un-encroached grasslands to mature, closed-canopy ERC forest. We did not expect this same pattern for *A. maculatum*, as it is known to be associated with grassland in rural contexts [29]. This study will contribute to the expanding field of landscape epidemiology by assessing the impact of ERC encroachment on disease ecology in a region facing a significant rise in vector-borne disease prevalence. The findings aim to inform strategies for managing ERC and reducing disease risks to human, wildlife, and livestock populations.

## Methods

### Study design and site selection

We sampled ticks in western and central Oklahoma, USA, where ERC appears to facilitate tick survival and distribution [24]. We first identified areas in this region that were experiencing large-scale ERC encroachment and contained varying

stages of encroachment ranging from open grasslands (which we considered as our control treatment) to closed-canopy ERC forests. Within these areas we sought to locate clusters of four study sites (each site >2 ha) that were (1) within ~5 km of each other, and (2) that each captured one of four encroachment stages described in detail below (including control grasslands with no ERC). We specifically focused on areas that were publicly accessible (e.g., state parks/wildlife management areas; properties owned by Oklahoma State University) and sought permission from each entity. We identified these general areas using ArcGIS 10.1 (Environmental Systems Research Institute, Redlands, California) and the Oklahoma Ecological Systems Mapping (OESM) data layer, a 10x10-m resolution land cover layer that covers all of Oklahoma and includes specific land cover types for which ERC is a dominant species [30]. Specifically, we considered candidate areas if they contained substantial coverage of OESM land cover categories that were defined in Diamond et al. (2015) as likely to capture open grasslands and eastern redcedar-encroached areas (see Table 1 for specific land cover categories used to identify each type of area).

For all candidate areas identified using the above process, we ground-truthed land cover types during field visits. Based on ground-truthing, we excluded areas that lacked adequate road access, that had different land covers than indicated in the OESM layer (including areas where ERC had been cut/removed), and that could not accommodate four sites capturing the stages of ERC encroachment described below (e.g., due to no or limited coverage of and/or no access to one or more ERC stages). For seven remaining areas, hereafter referred to as site clusters (Fig 2), we visually identified and georeferenced boundaries of four sites capturing distinct stages of ERC encroachment stages (i.e., 28 total sites, with four in each of the seven site clusters). These four ERC encroachment stages appear to be associated with varying tick abundances based on a pilot study [23,24]. These four stages included: (1) open grasslands with no ERC or other woody cover (hereafter, referred to as grassland sites); (2) grasslands with young ERC stands (4−6 yr old; 1- 2m tall) with scattered small trees and extensive intervening grass cover (hereafter, referred to as early ERC encroachment sites); (3) grasslands with medium-aged ERC stands (6−12 yr; 2-3m tall) with moderate encroachment by medium-sized trees and some intervening grass cover (hereafter, referred to as moderate ERC encroachment sites); and (4) mature ERC stands (≥12 yr; 4-5m tall) with large trees, closed canopies, and little to no grass remaining [31] (Fig 1) (hereafter, referred to as mature ERC encroachment sites). We note that these four stages refer, respectively, to stages 1, 2, 3, and 4, in a companion study of tick infestation of birds in the same study area [32]. Although we sought to separate all sites within clusters by at least 1 km, constraints related to logistics and areal coverage and location of encroachment stages resulted in some sites being as close as 0.11 km to each other. However, the average inter-site distance within clusters was much greater at 0.82 km (SE +/- 0.2 km). Further, as described under "Statistical Analyses" we accounted for spatial non-independence of sites within site clusters by treating cluster as a random effect in our generalized linear mixed models (GLMMs).

Regarding fire regimes at our study sites, we note that throughout our study region, frequent disturbance by fire, including wildfire and prescribed fire (i.e., recent prescribed fire efforts for rangeland, wildlife, and ecosystem management,

**Table 1. Specific land cover categories from the Oklahoma Ecological Systems Mapping (OESM) data layer (Diamond et al. 2015) used to identify areas in western and central Oklahoma, USA with open grasslands (which we considered to be our control treatment), and varying stages of eastern redcedar encroachment.**

| Land cover type | OESM land cover data layers used to identify each land cover type |
|---|---|
| Grassland | 207-Central Mixed grass Prairie: Prairie/Pasture |
| ERC encroached areas | 15005-Canyon: Juniper Shrubland; 2515-High Plains: Bottomland Eastern Redcedar Woodland and Shrubland; 2503-High Plains: Bottomland Hardwood – Eastern Redcedar Forest; 2715-High Plains: Riparian Eastern Redcedar Woodland and Shrubland; 2703-High Plains: Riparian Mixed Hardwood – Eastern Redcedar Woodland; 9115-Ruderal Eastern Redcedar Woodland and Shrubland; 14815-South Central Interior: Bottomland Eastern Redcedar Woodland and Shrubland; 15115-South Central Interior: Riparian Eastern Redcedar Woodland and Shrubland; 1815-Southeastern Great Plains: Bottomland Eastern Redcedar Woodland and Shrubland; 1915-Southeastern Great Plains; Riparian Eastern redcedar Woodland and Shrubland |

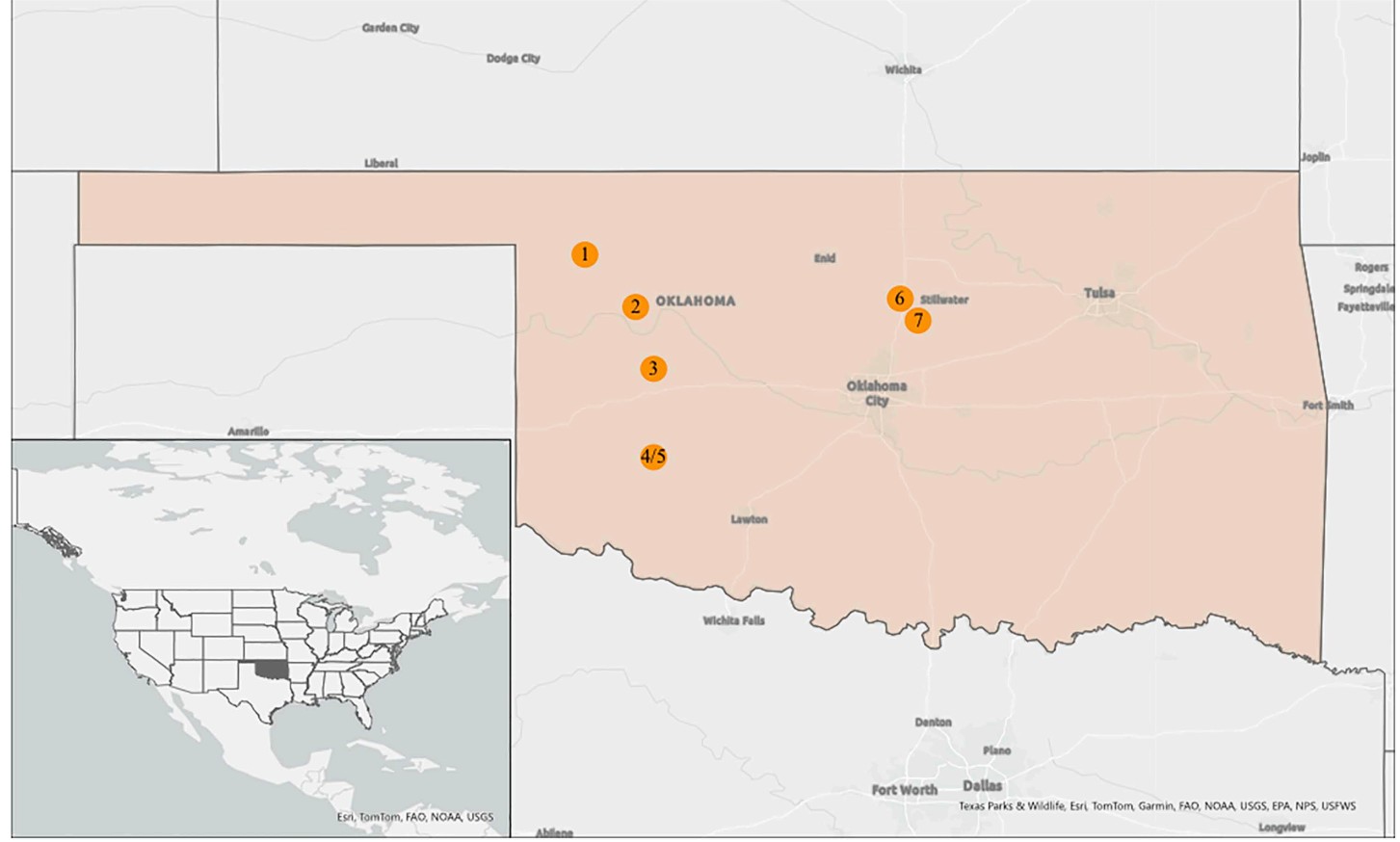

**Fig 2. Map of seven site clusters across central and western Oklahoma in which ticks were sampled to study the effect of woody plant encroachment by eastern redcedar on abundance of three tick species.** Sites included: 1) Boiling Springs State Park, 2) Canton Wildlife Management Area, 3) American Horse Lake, 4) Fort Cobb State Park West, 5) Fort Cobb State Park East, 6) Lake Carl Blackwell, 7) Oklahoma State University Research Range.

as well as historic use of fire by indigenous cultures for thousands of years) has historically maintained open grassland ecosystems [33,34]. Suppression of fire on the landscape has allowed expansion of tree cover, including encroachment by ERC. Detailed fire histories for some of our site clusters (e.g., those on lands managed by Oklahoma State University for research purposes) confirm that prescribed fire has regularly occurred at grassland sites and has been excluded from sites containing ERC trees. For site clusters without fire histories available, we infer that fires have likely occurred at grassland sites because otherwise these sites would likely have begun transitioning to ERC cover; likewise, at sites with ERC encroachment, we infer that fire—or at least top-killing canopy fires—have been excluded for at least for as long as ERC trees have been established.

## Tick collection

From July 2022 through June 2024, adult and nymphal ticks were collected at all 28 sites every three weeks during the peak period of activity for most tick species in Oklahoma (April 1st - July 15th), as well as once during early spring (March 2023 and 2024) and twice during late summer/fall (July 16th - Oct 31st 2022 and 2023), when tick activity is typically lower than in early summer. Tick collections were also conducted once in winter (Nov-Feb) of both 2022 and 2023 at the two central Oklahoma sites because ticks (especially *Ixodes scapularis*, the vector for Lyme disease) can be active in our

study area on sunny and/or warm days throughout the winter. Due to safety considerations related to hunting seasons at one site cluster located on an Oklahoma Department of Wildlife Conservation State Wildlife Management Area (WMA), collection schedules had to be adjusted slightly for the sites in this cluster. Specifically, the spring (May) and fall (October) collections at the WMA had to be conducted between the hours of 1000 and 1500 on weekdays only. All collections occurred on days with a temperature greater than 10°C and windspeeds below 32 kph, as these conditions are most suitable for tick collection. Collections took place between the hours of 0800 and 1200 hr during the summer months (June-August) because afternoon temperatures in Oklahoma frequently become too hot in the afternoon for tick collection. However, during the fall and winter months (October-February), and with the above exception for the WMA sites, collections took place in the afternoon between 1300 and 1700 hr, as this is the ideal time of day to collect *I. scapularis* [35]. Each round of collections at a site cluster consisted of placing three $CO_2$ traps at each of the four sites and flagging two 100m linear transects at each site following U.S. Centers for Disease Control (CDC) guidelines for tick flagging [36]. $CO_2$ traps consisted of placing a container of dry ice (solid $CO_2$) in the middle of a 0.6 x 0.6 m wooden board lined with wide masking tape [24]. All $CO_2$ traps were left open for approximately 1.5 hr. For sites with ERC trees (i.e., all sites except grassland sites), each $CO_2$ trap was placed under a different ERC tree; for grassland sites, each $CO_2$ trap was in open grassland. To prevent oversampling and emphasize analysis at the level of entire sites, $CO_2$ traps and flagging transects were randomly relocated each visit. Both trapping methods were also used in winter to ensure collection of as many active ticks as possible. Ticks were placed in vials with 70% EtOH by date, site, and collection method, and transported to the lab for identification using standard keys [37–39]. and stored at −20°C until processed for tick-borne pathogen testing.

## Tree cover and height measurements

To confirm that there were indeed differences in the stage of ERC encroachment among the four site categories that we used for site selection, we estimated and descriptively compared two characteristics of encroachment, average ERC height and horizontal percent cover of ERC, among the four encroachment stages based on measurements taken at all 28 sites in summer 2024. Measurements were taken at three representative locations within each site cluster, with one location at each ERC-encroached stage (early, moderate, and mature ERC stages); these measurements were not taken in grassland sites because they contained no ERC trees. For ERC height measurements at each site, one tree was selected in each of five randomly chosen compass directions (N, NE, E, SE, S, SW, W, and NW) that were randomly selected using an orientation generator; this design resulted in 15 total tree height measurements per site cluster (5 trees at each of the 3 ERC-encroached stages). Tree height was measured using a TruPulse laser hypsometer; we selected three points on the tree (horizontal angle from measurer, top, bottom) using a filter set for FARTHEST to prevent tree limbs from interfering with the measurement. For ERC horizontal percent cover, we laid out a 10m circular radius plot, centered on the $CO_2$ trap location, using a measuring tape and surveyor's flags placed at each of the four cardinal directions. For each quadrant within the circular plot, we visually estimated the percentage of the quadrant that was horizontally covered by living ERC trees. To avoid individual bias, the same individual conducted all measurements. All individual tree height and percent cover readings were averaged at the site-level.

## Statistical analysis

This study aimed to evaluate the effect of ERC encroachment on tick abundance in central and western Oklahoma. Because different life stages of *A. americanum* may respond to ERC differently [40], and because different sampling approaches target different tick behaviors and stages [41], we conducted four total analyses for *A. americanum*, including for abundance of: (1) adults collected by $CO_2$ trapping, (2) adults collected by flagging, (3) nymphs collected by $CO_2$ trapping, and (4) nymphs collected by flagging. However, for *D. variabilis* and *A. maculatum,* we only analyzed effects of ERC on total abundance due to these species being sampled in relatively low numbers, which limited our ability to conduct analyses for separate life stages and trapping methods. Analysis was conducted in R-4.2.3 [42].

For all of the above analyses, and because tick data were counts, we used generalized linear mixed models (GLMMs) and tested for overdispersion [43] to determine whether to use a Poisson or negative binomial distribution; Poisson and negative binomial models were respectively run in R packages lme4 [44] and MASS [45]. Overdispersion test results based on null (i.e., intercept only) models were statistically significant for all six analyses (all p-values<0.001 detecting overdispersion in Poisson models) indicating a need to use negative binomial models. Due to a high frequency of zero counts, we also used Akaike Information Criterion (AIC) model comparison to determine if zero-inflated negative binomial models were more appropriate. The ΔAIC value for the zero-inflated negative binomial model was always higher than for the negative binomial model; therefore, we used negative binomial GLMMs for all six analyses. Our unit of replication for all analyses was individual visits to each site (i.e., number of ticks sampled during each site visit, with 15 visits to each of the 28 sites resulting in 420 replicates). To account for multiple replicates/visits per site and for the greater similarity of sites within clusters compared to sites in different clusters, site nested within site cluster was included as a random effect in all analyses. To evaluate the effect of ERC encroachment stage on each tick abundance response variable, we used the above negative binomial GLMM structure and defined both a null (intercept-only) model and a model containing a fixed effect for stage (a categorical variable with four levels). We evaluated statistical significance of the effect of stage using p-values (α-level = 0.05), and for models with a significant effect of stage identified, we further compared differences in tick abundance among stages using the emmeans package [46] to perform post-hoc analyses, conducting pairwise comparisons among all combinations of ERC stages.

## Results

### Tree cover and height measurements

Descriptive comparisons of stage means for ERC tree height and horizontal percent cover confirmed that our sampling captured four distinct stages of encroachment. Specifically, both ERC height and horizontal percent cover increased with each successive ERC encroachment stage. Average ERC height was 0 m for grassland sites, 3.6 m for early ERC encroachment sites, 5.3 m for moderate ERC encroachment sites, and 7.2 m for mature ERC encroachment sites. Average horizontal cover of ERC for these four respective encroachment stages was 0%, 30.2%, 55.2%, and 89.3%. (S1 Fig).

### Summary of tick sampling results

A total of 44,833 ticks were collected from 7 site clusters across central and western Oklahoma between July 2022 and June 2024 (Table 2), with greater abundance in 2024 (26,490, 59%) than 2023 (18,343, 41%). The majority of ticks were *Amblyomma americanum* (44,296, 98.8%), followed by *Dermacentor variabilis* (360, 0.8%), *A. maculatum* (153, 0.3%), *Ixodes scapularis* (16, 0.04%), and *Haemaphysalis leporispalustris* (8 nymphs, 0.02%). Of 44,296 *A. americanum*, 16,073 (36%) were adults (7,669 males and 8,404 females) and 28,223 (64%) were nymphs. *A. americanum* and *D. variabilis* were found at every site cluster, while *A. maculatum* was found at 6 of 7 (86%) site clusters. The highest counts of *A. americanum* (20,422 total, including 14,975 nymphs and 5,447 adults) came from Research Range, a central Oklahoma site cluster, while the highest counts of both *D. variabilis* (184) and *A. maculatum* (56) came from Canton, a western Oklahoma site cluster. The highest count of *A. americanum* for a single collection visit came from an early ERC encroachment site in our Research Range site cluster that was sampled in May 2024 (11,868 individuals collected, accounting for 26.8% of the total *A. americanum* count, with 11,452 of these individuals from a single trap). To avoid model convergence issues and to prevent this data point from obscuring overarching patterns across our desired spatiotemporal scope of inference, all $CO_2$ traps for this site visit were excluded from our statistical analyses.

### Effects of Eastern redcedar encroachment stage on tick abundance

For *A. americanum* adults collected from $CO_2$ traps, there was a significant effect of ERC stage on abundance (F = 120.57, df = 3, p < 0.001). Pairwise comparisons of group means revealed that *A. americanum* abundance was

**Table 2. Total count (mean +/- SE) of field-collected ticks sampled by site, stage, species, and life stage at seven site clusters and 28 sites capturing open grasslands and three stages of intensifying eastern redcedar encroachment (respectively labelled in the table as open grass-lands, early ERC, moderate ERC, and mature ERC) across central and western Oklahoma, USA, July 2022-June 2024.**

| Site | Stage | N of Visits | *A. america-num* Adults (mean +/- SE) | *A. america-num* Nymphs (mean +/- SE) | *D. variabi-lis* Adults (mean +/- SE) | *A. macula-tum* Adults (mean +/- SE) | *A. macula-tum* Nymphs (mean +/- SE) | *I. scapu-laris* Adults (mean +/- SE) | *H. leporispal-ustris* Nymphs (mean +/- SE) |
|---|---|---|---|---|---|---|---|---|---|
| AHL[a] | Open grassland | 16 | 0 (0.0 +/- 0.0) | 0 (0.0 +/- 0.0) | 2 (0.1 +/- 0.1) | 1 (0.1 +/- 0.1) | 0 (0.0 +/- 0.0) | 0 (0.0 +/- 0.0) | 0 (0.0 +/- 0.0) |
| | Early ERC | 16 | 3 (0.2 +/- 0.1) | 4 (0.3 +/- 0.1) | 4 (0.3 +/- 0.1) | 0 (0.0 +/- 0.0) | 0 (0.0 +/- 0.0) | 0 (0.0 +/- 0.0) | 0 (0.0 +/- 0.0) |
| | Moderate ERC | 16 | 5 (0.3 +/- 0.2) | 5 (0.3 +/- 0.3) | 5 (0.3 +/- 0.2) | 0 (0.0 +/- 0.0) | 0 (0.0 +/- 0.0) | 0 (0.0 +/- 0.0) | 0 (0.0 +/- 0.0) |
| | Mature ERC | 16 | 10 (0.6 +/- 0.3) | 2 (0.1 +/- 0.1) | 6 (0.4 +/- 0.2) | 0 (0.0 +/- 0.0) | 0 (0.0 +/- 0.0) | 0 (0.0 +/- 0.0) | 0 (0.0 +/- 0.0) |
| BS[b] | Open grassland | 16 | 1 (0.1 +/- 0.1) | 4 (0.3 +/- 0.2) | 1 (0.1 +/- 0.1) | 0 (0.0 +/- 0.0) | 0 (0.0 +/- 0.0) | 0 (0.0 +/- 0.0) | 0 (0.0 +/- 0.0) |
| | Early ERC | 16 | 238 (14.9 +/- 6.0) | 848 (53.0 +/- 22.3) | 24 (1.5 +/- 0.5) | 0 (0.0 +/- 0.0) | 0 (0.0 +/- 0.0) | 0 (0.0 +/- 0.0) | 0 (0.0 +/- 0.0) |
| | Moderate ERC | 16 | 116 (7.3 +/- 2.5) | 80 (5.0 +/- 2.4) | 17 (1.1 +/- 0.3) | 0 (0.0 +/- 0.0) | 0 (0.0 +/- 0.0) | 0 (0.0 +/- 0.0) | 0 (0.0 +/- 0.0) |
| | Mature ERC | 16 | 405 (25.3 +/- 8.5) | 297 (18.6 +/- 6.1) | 15 (0.9 +/- 0.3) | 0 (0.0 +/- 0.0) | 0 (0.0 +/- 0.0) | 0 (0.0 +/- 0.0) | 2 (0.1 +/- 0.1) |
| CA[c] | Open grassland | 16 | 14 (0.9 +/- 0.4) | 24 (1.5 +/- 0.8) | 5 (0.3 +/- 0.1) | 19 (1.2 +/- 0.4) | 0 (0.0 +/- 0.0) | 0 (0.0 +/- 0.0) | 0 (0.0 +/- 0.0) |
| | Early ERC | 16 | 212 (13.3 +/- 3.1) | 774 (48.4 +/- 14.5) | 61 (3.8 +/- 1.4) | 18 (1.1 +/- 0.4) | 0 (0.0 +/- 0.0) | 2 (0.1 +/- 0.1) | 1 (0.1 +/- 0.1) |
| | Moderate ERC | 16 | 327 (20.4 +/- 5.7) | 499 (31.2 +/- 9.1) | 73 (4.6 +/- 1.4) | 15 (0.9 +/- 0.3) | 0 (0.0 +/- 0.0) | 0 (0.0 +/- 0.0) | 0 (0.0 +/- 0.0) |
| | Mature ERC | 16 | 129 (8.1 +/- 3.0) | 68 (4.3 +/- 1.3) | 45 (2.8 +/- 0.9) | 3 (0.2 +/- 0.1) | 1 (0.1 +/- 0.1) | 0 (0.0 +/- 0.0) | 1 (0.1 +/- 0.1) |
| CB[d] | Open grassland | 16 | 17 (1.1 +/- 0.3) | 19 (1.2 +/- 0.6) | 6 (0.4 +/- 0.2) | 4 (0.3 +/- 0.1) | 0 (0.0 +/- 0.0) | 0 (0.0 +/- 0.0) | 0 (0.0 +/- 0.0) |
| | Early ERC | 16 | 1,866 (116.6 +/- 36.2) | 2,899 (181.2 +/- 51.1) | 11 (0.7 +/- 0.2) | 0 (0.0 +/- 0.0) | 0 (0.0 +/- 0.0) | 0 (0.0 +/- 0.0) | 0 (0.0 +/- 0.0) |
| | Moderate ERC | 16 | 1,005 (62.8 +/- 15.1) | 1,523 (95.2 +/- 35.5) | 3 (0.2 +/- 0.1) | 1 (0.1 +/- 0.1) | 0 (0.0 +/- 0.0) | 2 (0.1 +/- 0.1) | 0 (0.0 +/- 0.0) |
| | Mature ERC | 16 | 961 (60.1 +/- 11.3) | 1,382 (86.4 +/- 20.4) | 11 (0.7 +/- 0.3) | 1 (0.1 +/- 0.1) | 0 (0.0 +/- 0.0) | 5 (0.3 +/- 0.2) | 1 (0.1 +/- 0.1) |
| RR[e] | Open grassland | 16 | 25 (1.6 +/- 0.7) | 53 (3.3 +/- 1.6) | 1 (0.1 +/- 0.1) | 0 (0.0 +/- 0.0) | 0 (0.0 +/- 0.0) | 0 (0.0 +/- 0.0) | 0 (0.0 +/- 0.0) |
| | Early ERC | 16 | 3,848 (240.5 +/- 124.8) | 12,119 (757.4 +/- 669.5) | 2 (0.1 +/- 0.1) | 11 (0.7 +/- 0.3) | 0 (0.0 +/- 0.0) | 2 (0.1 +/- 0.1) | 0 (0.0 +/- 0.0) |
| | Moderate ERC | 16 | 1,134 (70.9 +/- 23.3) | 1,278 (79.9 +/- 39.1) | 3 (0.2 +/- 0.1) | 0 (0.0 +/- 0.0) | 0 (0.0 +/- 0.0) | 1 (0.1 +/- 0.1) | 0 (0.0 +/- 0.0) |
| | Mature ERC | 16 | 440 (27.5 +/- 5.7) | 1,525 (95.3 +/- 30.5) | 1 (0.1 +/- 0.1) | 0 (0.0 +/- 0.0) | 0 (0.0 +/- 0.0) | 0 (0.0 +/- 0.0) | 2 (0.1 +/- 0.1) |
| FCE[f] | Open grassland | 16 | 87 (5.4 +/- 1.6) | 474 (29.6 +/- 17.8) | 0 (0.0 +/- 0.0) | 39 (2.4 +/- 1.0) | 0 (0.0 +/- 0.0) | 2 (0.1 +/- 0.1) | 0 (0.0 +/- 0.0) |
| | Early ERC | 16 | 439 (27.4 +/- 8.7) | 462 (28.9 +/- 13.6) | 2 (0.1 +/- 0.1) | 0 (0.0 +/- 0.0) | 0 (0.0 +/- 0.0) | 1 (0.1 +/- 0.1) | 0 (0.0 +/- 0.0) |
| | Moderate ERC | 16 | 699 (43.7 +/- 9.1) | 240 (15.0 +/- 4.5) | 11 (0.7 +/- 0.2) | 0 (0.0 +/- 0.0) | 1 (0.1 +/- 0.1) | 0 (0.0 +/- 0.0) | 0 (0.0 +/- 0.0) |

*(Continued)*

**Table 2.** (Continued)

| Site | Stage | N of Visits | *A. americanum* Adults (mean +/- SE) | *A. americanum* Nymphs (mean +/- SE) | *D. variabilis* Adults (mean +/- SE) | *A. maculatum* Adults (mean +/- SE) | *A. maculatum* Nymphs (mean +/- SE) | *I. scapularis* Adults (mean +/- SE) | *H. leporispalustris* Nymphs (mean +/- SE) |
|---|---|---|---|---|---|---|---|---|---|
| | Mature ERC | 16 | 572 (35.8 +/- 13.2) | 194 (12.1 +/- 2.3) | 11 (0.7 +/- 0.3) | 1 (0.1 +/- 0.1) | 0 (0.0 +/- 0.0) | 0 (0.0 +/- 0.0) | 0 (0.0 +/- 0.0) |
| FCW[g] | Open grassland | 16 | 26 (1.6 +/- 0.6) | 28 (1.8 +/- 0.5) | 0 (0.0 +/- 0.0) | 23 (1.4 +/- 0.7) | 0 (0.0 +/- 0.0) | 0 (0.0 +/- 0.0) | 0 (0.0 +/- 0.0) |
| | Early ERC | 16 | 1,457 (91.1 +/- 37.0) | 1,791 (111.9 +/- 50.5) | 6 (0.4 +/- 0.2) | 1 (0.1 +/- 0.1) | 0 (0.0 +/- 0.0) | 0 (0.0 +/- 0.0) | 0 (0.0 +/- 0.0) |
| | Moderate ERC | 16 | 576 (36.0 +/- 7.4) | 1,069 (66.8 +/- 21.1) | 17 (1.1 +/- 0.4) | 10 (0.6 +/- 0.3) | 0 (0.0 +/- 0.0) | 1 (0.1 +/- 0.1) | 1 (0.1 +/- 0.1) |
| | Mature ERC | 16 | 1,461 (91.3 +/- 55.6) | 562 (35.1 +/- 11.6) | 17 (1.1 +/- 0.3) | 4 (0.3 +/- 0.2) | 0 (0.0 +/- 0.0) | 0 (0.0 +/- 0.0) | 0 (0.0 +/- 0.0) |
| | Total | | 16,073 | 28,223 | 360 | 151 | 2 | 16 | 8 |

[a]American Horse Lake [b]Boiling Springs State Park [c]Canton Wildlife Management Area [d]Lake Carl Blackwell [e]Oklahoma State University Research Range [f]Fort Cobb State Park East [g]Fort Cobb State Park West

significantly higher in early, moderate, and mature ERC encroachment sites than in grasslands (p-value for all stage comparisons with grassland < 0.001). However, there were no significant differences in abundance between the three stages with ERC encroachment present (early ERC:moderate ERC: p = 0.1027; early ERC:mature ERC: p = 0.5574; moderate ERC:mature ERC: p = 0.7999) (Fig 3a). Based on estimated stage means, there were approximately 66.2 times more *A. americanum* adults collected by $CO_2$ traps in early ERC encroachment sites compared to open grassland sites.

For *A. americanum* adults collected from flags, there was a significant effect of ERC stage on abundance (F = 16.12, df = 3, p < 0.001). Pairwise comparisons of group means revealed that *A. americanum* abundance was significantly higher in early, moderate, and mature ERC encroachment sites than in grasslands (p-value for all stage comparisons with grassland < 0.001). However, there were no significant differences in abundance between the three stages with ERC encroachment present (early ERC:moderate ERC: p = 0.9924; early ERC:mature ERC: p = 0.9063; moderate ERC:mature ERC: p = 0.9803) (Fig 3b). Based on estimated stage means, there were approximately 5.9 times more *A. americanum* adults collected by flagging in early ERC encroachment sites compared to open grassland sites.

For *A. americanum* nymphs collected from $CO_2$ traps, there was a significant effect of ERC stage on abundance (F = 68.73, df = 3, p < 0.001). Pairwise comparisons of group means revealed that *A. americanum* abundance was significantly higher in early, moderate, and mature ERC encroachment sites than in grasslands (p-value for all stage comparisons with grassland < 0.001). There also were significant differences in abundance between the three stages with ERC encroachment present, with early ERC sites having significantly more ticks than moderate ERC and mature ERC sites (early ERC:moderate ERC: p = 0.0028; early ERC:mature ERC: p < 0.001) but comparable abundance between moderate ERC and mature ERC sites (p = 0.7007) (Fig 3c). Based on estimated stage means, there were approximately 20.6 times more *A. americanum* nymphs collected by $CO_2$ traps in early ERC encroachment sites compared to open grassland sites.

For *A. americanum* nymphs collected through flagging, there was a significant effect of ERC stage on abundance (F = 9.29, df = 3, p < 0.001). Pairwise comparisons of group means revealed that *A. americanum* abundance was significantly higher in early ERC and mature ERC sites than in both grassland sites (open grassland:early ERC: p = 0.0026; open grassland:mature ERC: p = 0.0004) and moderate ERC sites (early ERC:moderate ERC: p = 0.0433; moderate ERC:mature ERC: p = 0.0102). However, abundance was comparable between early ERC and mature ERC sites (p = 0.9756) and between open grassland and moderate ERC sites (p = 0.6399) (Fig 3d). Based on estimated stage means, there were approximately 2.8 times more *A. Americanum* nymphs collected by flagging in early ERC encroachment sites compared to open grassland sites.

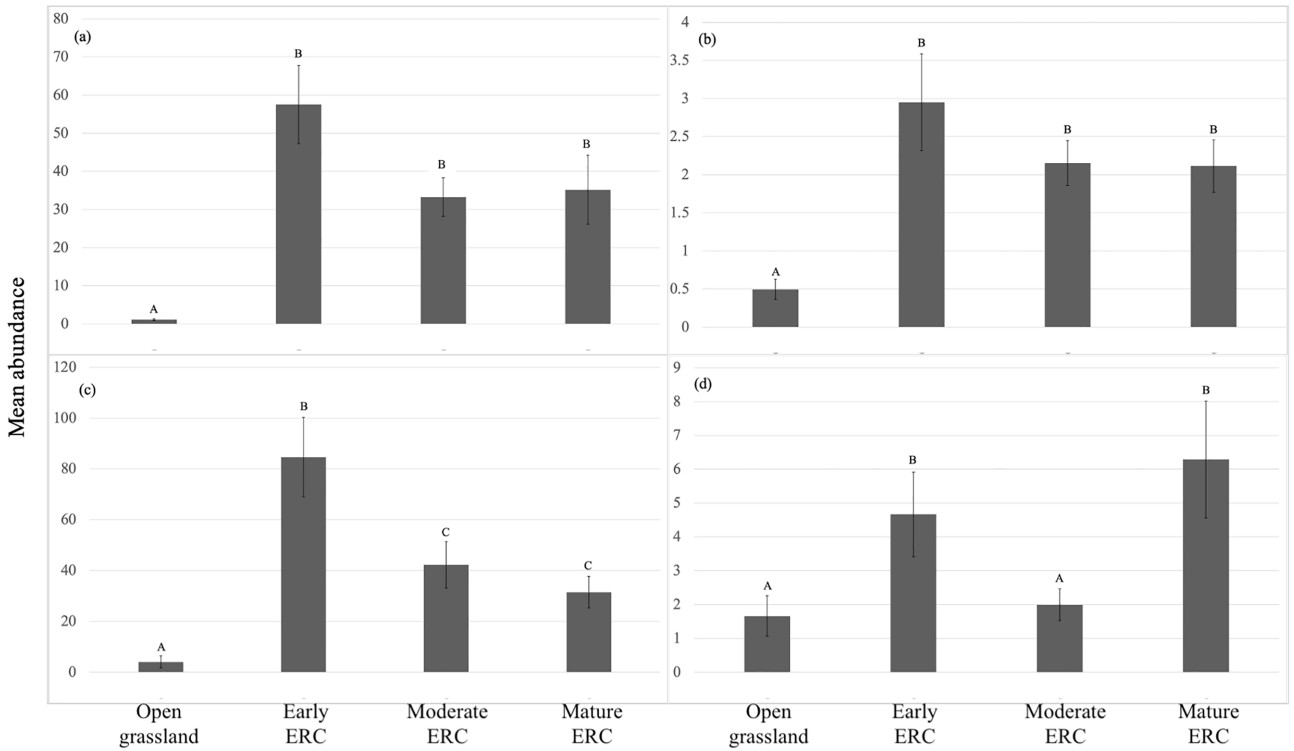

**Fig 3. Comparison of mean abundances of: (a)** *A. americanum* **adults collected from $CO_2$ traps, (b)** *A. americanum* **adults collected from flags, (c)** *A. americanum* **nymphs collected from $CO_2$ traps, (d)** *A. americanum* **nymphs collected from flags, among sites capturing open grasslands, early eastern redcedar (ERC), moderate ERC, and mature ERC.** Whiskers represent standard errors, letters shared among stages indicate no significant differences among them, and stages with different letters indicate significant differences among them.

For total abundance of *D. variabilis*, there was a significant effect of ERC stage on abundance (F = 14.00, df = 3, p < 0.001). Pairwise comparisons of group means revealed that *D. variabilis* abundance was significantly higher in early, moderate, and mature ERC encroachment sites than in grasslands (p-value for all stage comparisons with grassland < 0.001). However, there were no significant differences in abundance between the three stages with ERC encroachment present (early ERC:moderate ERC: p = 0.8238; early ERC:mature ERC: p = 0.9657; moderate ERC:mature ERC: p = 0.9805) (Fig 4). Based on estimated stage means, there were approximately 7.5 times more *D. variablis* in early ERC encroachment sites compared to open grassland sites.

For total abundance of *A. maculatum*, there was a significant effect of ERC stage on abundance (F = 10.31, df = 3, p < 0.001). Pairwise comparisons of group means revealed that *A. maculatum* abundance was significantly higher in open grassland than early, moderate, and mature ERC encroachment sites (open grassland:early ERC: p = 0.0227; open grassland:moderate ERC: p = 0.0018; open grassland:mature ERC p < 0.001). However, there were no significant differences in abundance between the three stages with ERC encroachment present (early ERC:moderate ERC: p = 0.9207; early ERC:maturre ERC: p = 0.0774; moderate ERC:mature ERC: p = 0.2334) (Fig 5). Based on estimated stage means, there were on average approximately 2.8 times more *A. maculatum* in open grassland sites compared to early ERC encroachment sites.

## Discussion

Results from this study identified that ERC encroachment in the U.S. Great Plains likely increases the abundance of two tick species, while abundance of another species known to be associated with grassland ecosystems was lower in

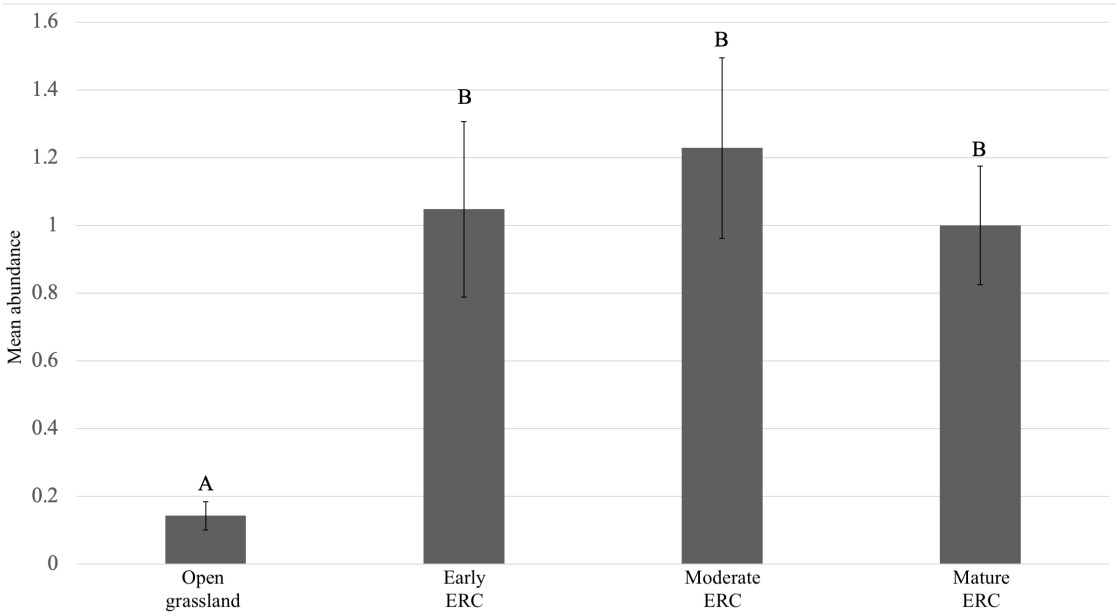

**Fig 4. Comparison of mean abundance of *D. variabilis* among sites capturing open grasslands, early eastern redcedar (ERC), moderate ERC, and mature ERC.** Whiskers represent standard errors, letters shared among stages indicate no significant differences among them, and stages with different letters indicate significant differences among them.

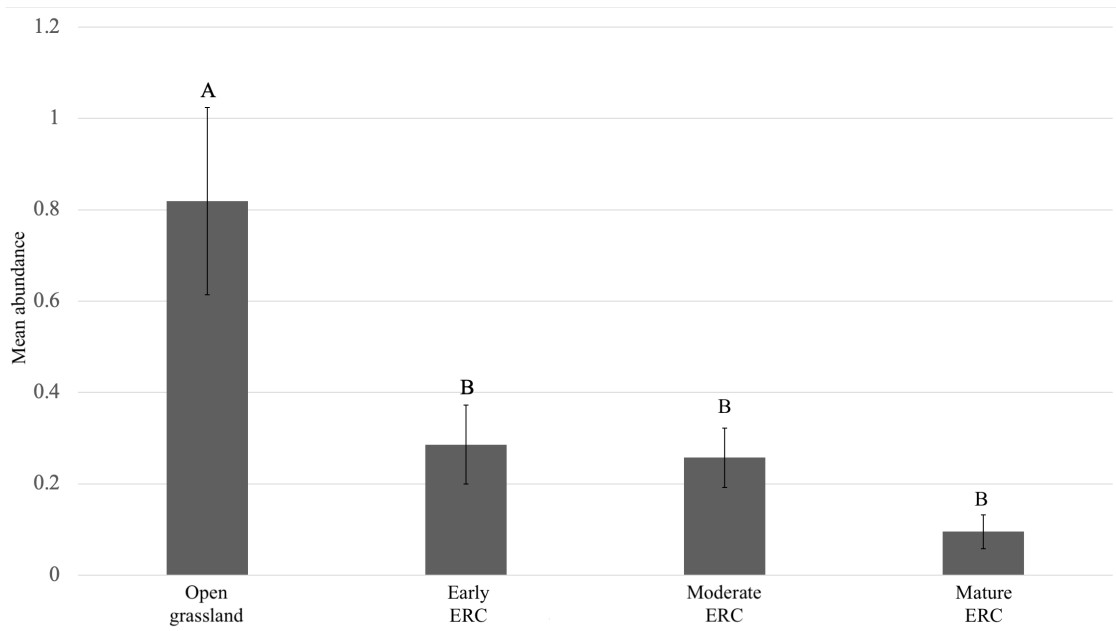

**Fig 5. Comparison of mean abundance of *A. maculatum* among sites capturing open grasslands, early eastern redcedar (ERC), moderate ERC, and mature ERC.** Whiskers represent standard errors, letters shared among stages indicate no significant differences among them, and stages with different letters indicate significant differences among them.

ERC-encroached areas compared to open grasslands. Notably, for all analyses involving *A. americanum* and *D. variabilis*, the earliest stage of ERC encroachment had significantly higher tick abundance compared to open grasslands, indicating that abundances of these tick species increase shortly after the onset of ERC encroachment into grasslands. We also documented some species- and life stage-specific responses to later stages of ERC encroachment. For adult *A. americanum* collected from flagging and $CO_2$ traps, and *D. variabilis* collected from flagging and $CO_2$ traps, abundances were similarly high between early, moderate, and mature stages of ERC encroachment. For A. *americanum* nymphs collected by flagging, abundances in moderate ERC encroachment sites were lower than in early ERC encroachment sites but increased again in mature ERC encroachment sites to high levels that were comparable to early ERC sites. For *A. americanum* nymphs collected by $CO_2$ traps, abundance was significantly higher in early ERC encroachment sites than in both moderate and mature ERC encroachment sites. For *A. maculatum*, a species known to be associated with open environments, abundance was higher in grassland compared to all 3 stages with ERC encroachment. Past research has shown that mature, closed canopy stands of ERC facilitate increased *A. americanum* and *D. variabilis* abundance under individual ERC trees [24,25], however, this is the first study to demonstrate more nuanced responses of tick abundance to intensifying levels of encroachment on a broader, site-level scale.

Contrary to our hypothesis, abundances of *A. americanum* and *D. variabilis* did not increase with each successive stage of ERC encroachment. Abundances for these species were always significantly higher in the earliest stage of ERC encroachment compared to grassland; however, as described above and with the exception of *A. americanum* nymphs collected by flagging and $CO_2$ traps, abundance was generally equivalent across early, moderate, and mature stages of ERC encroachment. Variation in ground layer vegetation (e.g., grasses and forbs) among encroachment stages may explain the similarity in tick abundances among sites with ERC trees that are experiencing different stages of invasion. Previous research has reported that reduced ground cover results in lower *A. americanum* abundance [40], while *D. variabilis* is typically found in field-forest ecotones, which are characterized by a dense understory [47]. We did not conduct vegetation sampling in our sites; however, increasing cover of ERC is known to reduce understory vegetation cover and biomass [48–50]. We also observed anecdotally that sites with mature ERC trees had sparse ground-layer vegetation due to the extensive tree canopy that blocked light penetration to the forest floor, which was largely comprised of ERC needles, fallen limbs, and bare soil. In contrast, early and moderate ERC encroachment sites featured native grasses like Buffalo grass (*Bouteloua dactyloides*), Bluestem (genus *Andropogon*), and Switchgrass (*Panicum virgatum*), along with additional woody plants such as Smooth Sumac (*Rhus glabra*) (Propst, personal observation). Increasing ERC cover likely changes abiotic conditions (e.g., increasing humidity and reducing wind velocity [20]) in ways that support increasing tick abundance; however, reduced ground cover associated with later encroachment stages may offset effects of increasing tree cover, leading to the pattern we observed of tick abundances being equally high among early, moderate, and mature stages of ERC encroachment.

Tick distribution and population establishment are dependent upon wildlife host availability and abundance [51], and responses of wildlife to ERC encroachment may also help explain these results. White-tailed deer is the primary vertebrate host for *A. americanum*, though wild turkey (*Meleagris gallopavo*) and raccoon (*Procyon lotor*) also are known to contribute to maintenance of *A. americanum* populations in the United States [52,53]. All three of these host species are native to Oklahoma and may play a role in the westward spread of ticks, aligning with the expansion of ERC encroachment across the state [23]. Several bird species also carry a large number of ticks in our study region [54], and bird community changes in association with ERC encroachment are well-documented [55]. However, beyond a general understanding that wildlife species composition and abundance of individual species change with the process of WPE, including by ERC [55–57], there is limited understanding of more nuanced changes in wildlife abundance and behavior that may occur with intensifying stages of encroachment. These changes in wildlife-related factors likely overlay with other critical ERC-caused changes in abiotic conditions, understory vegetation, and leaf litter to influence tick abundance. Further research focused on use of different stages of ERC encroachment by vertebrate hosts, and on sampling of ticks from

various host species including birds and mammals, is needed to identify which hosts are utilized by each species and life-stage of ticks in different stages of ERC encroachment.

Our study used two collection methods, $CO_2$ trapping and flagging, to assess tick abundance and gain insights into tick host-seeking activity. Ticks collected by flagging offer a clear representation of actively host-seeking individuals in the environment, a critical phase during which humans are most likely to encounter and acquire ticks [36,58]. We found more active host-seeking *A. americanum* nymphs in ERC-encroached grasslands (specifically in early and mature ERC encroachment sites) compared to open grassland. Activities like hunting, ranching, and farming in ERC-encroached areas increase the risk of encountering host-seeking *A. americanum.* Given that our ERC site clusters were located on publicly accessible lands, including state parks and wildlife management areas, we are aware that the public is actively using these areas for outdoor recreation and hunting which increases their risk of exposure to tick-borne diseases. Although flagging provides an estimate of host-seeking ticks within a specific area, $CO_2$ trapping can be used to estimate *A. americanum* populations on a landscape level [59]. A previous study that used $CO_2$ trapping and mark/recapture protocols reported that one adult *A. americanum* captured per hour using dry ice ($CO_2$) represented a mean estimation of 397 ticks per ha [59]. By applying this calculation to our data, we estimate the mean per ha *A. americanum* abundances for open grasslands, and early, moderate, and mature ERC encroachment sites to be 291.1, 15,226.3, 8,797.5, and 9,305.7, ticks per ha respectively. Although our formal analyses focused on identifying statistically significant differences among ERC encroachment stages, these descriptive estimates again illustrate the pattern shown for *A. americanum* adults, and for *D. variabilis,* that any level of encroachment may lead to higher numbers of ticks compared to grasslands. These estimates also highlight the possibility that the earliest stage of encroachment may be associated with the highest tick abundance. These broad-scale estimates of tick density suggest that it is critical to prevent ERC encroachment (e.g., through management methods outlined below) even before it reaches its earliest stage (i.e., during or prior to seed establishment) [34].

A single $CO_2$ trap in one of our early ERC encroachment sites collected 11,452 *A. americanum.* This is unusual and, to our knowledge, may be the largest reported number of ticks ever collected from one trap. This $CO_2$ trap was unique in that the collection occurred in central Oklahoma in late May 2024 just after a heavy rain event that caused flooding (pooling of water) near the trap location. Although we cannot completely rule out the possibility that these sampled ticks were always present in the vicinity of the trapping location, it seems possible that they may have congregated and accumulated there from a much larger area as a result of the flooding. Fire ants are well known for their short-term 'rafting' behaviors during flood conditions [60] but there is a gap in our knowledge regarding how ticks deal with flooding with more published studies involving underwater survival [61]. Laboratory studies focused on the underwater survival of ticks have reported that *A. americanum* can survive as long as 70 days under freshwater [62,63] but no studies have evaluated what ticks do in water when subjected to pooling or flooding conditions. The collection of such a large number of *A. americanum* from a single $CO_2$ trap suggests that ERC may provide favorable microhabitats for ticks to shelter in during specific seasons and adverse weather conditions. If this is indeed the case, the formation of large clusters of ERC, particularly in areas prone to flooding, may increase the risk of tick-borne pathogen transmission to wildlife or livestock seeking cover under ERC stands. More studies are needed to investigate how effects of ERC encroachment on ticks may vary in association with seasonality, with variation in short term weather including temperature and precipitation extremes, and with differences in climate including spatial differences between regions and temporal changes associated with climate change.

As with all studies, our research was characterized by several limitations. One potential limitation involved the placement of $CO_2$ traps beneath individual ERC trees in early, moderate, and mature ERC encroachment sites, which may have contributed to higher observed abundances than if we had also placed traps in patches of open grass between ERC trees. In the early stage of encroachment, ERC trees are large enough to provide suitable tick habitat, but as they remain relatively isolated from other trees, the individual trees may serve as islands of favorable tick habitat within less-favorable grass patches [25]. In contrast, areas experiencing moderate and mature ERC encroachment may feature more expansive tick habitats due to the presence of larger ERC trees that are closer together and have smaller intervening

grass patches. These characteristics may allow ticks to disperse more widely, possibly resulting in lower densities and fewer being captured under individual trees. Future research within each stage that compares tick abundances between ERC and grass patches would address the potential clustering effect and provide clearer insights into both fine-scale and site-level tick densities and abundances. Another potential limitation was that we only used seven site clusters for the study, which limited our site-level replication. While we tried to identify additional site clusters throughout the study area, we were constrained by the lack of public land in Oklahoma and by the difficulty of acquiring permission to assess private land. As such, we were only able to gain access to seven site clusters on land that was either open to the public or owned by Oklahoma State University. Finally, we did not evaluate seasonal variation in the relationship between ERC and tick abundance because our goal in this study was to generate information about the effect of ERC on ticks across the entire seasonal period of activity for each species. However, future research that tests hypotheses related to seasonal effects— for example, the possibility that differences in tick abundance between open grasslands and ERC-encroached areas are greatest during extreme conditions (e.g., hot and dry conditions in mid-to late summer) that are buffered by tree cover—is needed to increase understanding of the effects of woody plant encroachment on tick populations.

## Conclusion

Our results provide the most comprehensive evidence to date that ERC encroachment facilitates increased abundance of some tick species, specifically *A. americanum* and *D. variabilis*, compared to open grasslands not yet experiencing encroachment. Our analyses indicate that the earliest encroachment stage supports substantially greater tick abundance compared to open grasslands. This finding indicates that tick abundance increases shortly after the onset of encroachment and suggests that the most effective way to prevent increased exposure to ticks in grasslands is to prevent ERC encroachment before it starts or to remove it during its early stages [34].

Historically, the Great Plains was a grass-dominated biome with minimal tree cover and frequent disturbance by both wildfire and prescribed fire—including recent prescribed fire efforts and use of fire by indigenous cultures for thousands of years—however, fire suppression has facilitated expansion of woody plants, transforming grasslands into woodland ecosystems [34]. Restoring fire regimens is already known to be an effective approach to stop and reverse early stages of ERC, and our study indicates that this management strategy also may prevent tick populations from further expanding in association with this pervasive form of land cover change. Mechanical and herbicidal removal of ERC, though often more expensive, can also be used, including at later stages of encroachment when use of fire may become less effective and more dangerous [33]. Prescribed burns may lower tick numbers through mechanisms such as heat-induced mortality, reduced availability of shelter and questing areas, and modifications of microclimates coupled with a decrease in host availability [64]. A prior study in central Oklahoma that occurred on a property that included one of our site clusters reported that use of prescribed fire significantly reduced tick burdens on cattle but did not lower tick abundance within pastures [65]. However, other researchers have demonstrated that long-term prescribed burning can significantly reduce *A. americanum* populations [40,66]. Regardless, our results documenting increased populations of *A. americanum* and *D. variablis* at the onset of ERC encroachment provide further rationale for preventative management, since this approach should greatly reduce harmful impacts of encroachment on public and veterinary health outcomes, in addition to preventing adverse effects of ERC on biodiversity, water quantity, wildfire risk, and livestock production [34,67].

## Supporting information

**S1 Fig. Comparison of height and horizontal percent cover of eastern redcedar (ERC) trees in sites capturing open grasslands and three stages of intensifying ERC encroachment (referred to as early, moderate, and mature ERC encroachment sites here and in the text). For grasslands, no measurements of trees were taken because grassland sites contained no trees and therefore height and cover values were inferred to be 0. Horizontal lines**

in boxes indicate median values, boxes indicate interquartile ranges, and lines and whiskers indicate 95% confidence intervals.
(TIFF)

**S1 File. Abundance data for *A. americanum* adults collected from $CO_2$ traps, used for statistical analyses presented in this study.**
(XLSX)

**S2 File. Abundance data for *A. americanum* adults collected from flags, used for statistical analyses presented in this study.**
(XLSX)

**S3 File. Abundance data for *A. americanum* nymphs collected from $CO_2$ traps, used for statistical analyses presented in this study**
(XLSX)

**S4 File. Abundance data for *A. americanum* nymphs collected from flags, used for statistical analyses presented in this study.**
(XLSX)

**S5 File. Abundance data for *A. maculatum*, used for statistical analyses presented in this study.**
(XLSX)

**S6 File. Abundance data for *D. variabilis*, used for statistical analyses presented in this study.**
(XLSX)

## Acknowledgments

We would like to thank Meg Gilliland, Irene Eubanks, Olivia Horton, Billy Harsha, Emma Patterson, Presley Brandt, Mason Avelar, and Ryley Hall for their help collecting and processing data.

## Author contributions

**Conceptualization:** Jozlyn Propst, Tucker Taylor, Scott R. Loss, Bruce H. Noden.

**Data curation:** Jozlyn Propst.

**Formal analysis:** Jozlyn Propst.

**Funding acquisition:** Scott R. Loss, Bruce H. Noden.

**Investigation:** Jozlyn Propst, Tucker Taylor, Bruce H. Noden.

**Methodology:** Jozlyn Propst, Tucker Taylor.

**Supervision:** Scott R. Loss, Bruce H. Noden.

**Validation:** Scott R. Loss.

**Writing – original draft:** Jozlyn Propst.

**Writing – review & editing:** Tucker Taylor, Scott R. Loss, Bruce H. Noden.

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
