## [Decision Letter · Decision Letter 0]

14 May 2025

PONE-D-25-16397Effects of woody plant encroachment on abundance of multiple tick species in the U.S. Great PlainsPLOS ONE

Dear Dr. Propst,

Thank you for submitting your manuscript to PLOS ONE. After careful consideration, we feel that it has merit but does not fully meet PLOS ONE’s publication criteria as it currently stands. Therefore, we invite you to submit a revised version of the manuscript that addresses the points raised during the review process.

We look forward to receiving your revised manuscript.

Kind regards,

Xiao Guo, Ph.D.

Academic Editor

PLOS ONE

Journal Requirements:

4. We note that Figure 2 in your submission contain [map/satellite] images which may be copyrighted. All PLOS content is published under the Creative Commons Attribution License (CC BY 4.0), which means that the manuscript, images, and Supporting Information files will be freely available online, and any third party is permitted to access, download, copy, distribute, and use these materials in any way, even commercially, with proper attribution. For these reasons, we cannot publish previously copyrighted maps or satellite images created using proprietary data, such as Google software (Google Maps, Street View, and Earth). For more information, see our copyright guidelines: http://journals.plos.org/plosone/s/licenses-and-copyright.

Natural Earth (public domain):

Additional Editor Comments :

This manuscript has already been reviewed by two highly qualified experts, who have provided valuable feedback. Both reviewers recognize the potential of this study and find it appealing to the journal’s readership. At the same time, they have pointed out several issues, particularly regarding the need for more detailed descriptions in the Methods section and suggested revisions to the statistical analyses. Changes to the statistical methods, if implemented, may influence the study's results and subsequent discussion. I strongly recommend that the authors carefully address all of the reviewers’ comments and suggestions. The revised manuscript will be sent back to these expert reviewers for further evaluation. Whether the article will ultimately be accepted depends on the quality and completeness of these revisions.

Reviewers' comments:

Reviewer's Responses to Questions

**Comments to the Author**

1. Is the manuscript technically sound, and do the data support the conclusions?

Reviewer #1: Yes

Reviewer #2: Yes

2. Has the statistical analysis been performed appropriately and rigorously? 

Reviewer #1: No

Reviewer #2: Yes

3. Have the authors made all data underlying the findings in their manuscript fully available?

Reviewer #1: Yes

Reviewer #2: Yes

4. Is the manuscript presented in an intelligible fashion and written in standard English?

Reviewer #1: Yes

Reviewer #2: Yes

5. Review Comments to the Author

Reviewer #1: This manuscript presents an interesting study on tick abundance across different stages of eastern redcedar (ERC) encroachment. I offer several suggestions to improve clarity and analytical rigor in the methods, statistical analysis, and results presentation. Key concerns include justifying the use of abundance over density, clarifying how seasonal variation was handled, and adopting more appropriate statistical tools such as the appropriateness of using AIC and model comparisons when there are no competing models except the null, using emmeans for post-hoc comparisons and interpretation of model outputs. I recommend using more descriptive labels for categorical variables (e.g., "early ERC" instead of "stage 2") and translating beta coefficients into meaningful effect sizes (e.g., percent changes) for better reader comprehension. Minor comments include consistency in formatting, spelling, terminology, and presentation of duplicated information.

Methods

Statistical Analysis.

Is there a reason for using abundance for the tick drags rather than density of ticks?

How are you accounting for the different times of the year that ticks are collected in your analysis?

Regarding AIC, do you low sample sizes, it might be more appropriate to report AICc, which corrects for bias associated with small sample sizes and is more accurate.

Line 247, how are you defining greater support. Is this delta AIC > 2?

Line 249-254: At first read, I wasn’t sure what you mean by stage. I have to reread. Could you use the actual category (grassland, early ERC, moderate ERC and mature ERC. This would make the job of the reader easier. Also you are considering the grassland as the the reference stage or the control. I’m not sure what are you running separate models with each of the other stages of ERC encroachment as the reference for the intercept. To perform post comparisons, it would be more appropriate to use emmeans package as this would adjust the p-values for multiple comparisions?

Results

Table 2: I’m sure what you are using AIC when there are only two models to compare, one of which is the null. Usually you would have multiple predictors that you compare. In doing this, it would make more sense to just report the p-values.

Figures: for the X axis, instage of putting Stage 1,2,3 can you use “Open grassland”, “early ERC”, etc.

When describing the results, can you put it in context, instead of just stating the beta values, transform this into a percentage change. For instance in lines 318-319, based on the beta values, and if you are using a log link, the beta value can be convert indicates that "A. americanum was about 5 times as abundant in stage 2 as in stage 1," or ~400% more abundant, relative to stage 1."

I’m not sure how you got the letter values for the figures. What is a coefficient analysis? See above comments about using emmeans.

Throughout the manuscript, you have a tendency to have the same information (AIC values, beta values) in the text that is in a table. Choose one,

Discussion.

The discussion is well written given the current results, I won’t provide speciific comments as it is unknown if it will change when the method/results change.

Minor Comments

Line 168L misspelling “centeral”

Lines 179: be consistent with the date format, earlier, you put the month day and on this line you put day month.

Be consistent with numbers, when using numbers less than 10, spell out the entire word.

Line 258-263: what there an actually analysis for this? Also, the tect could be a bit more concise “Measurements of ERC trees confirmed that our sampling captured distinct stages of encroachment: both tree height and horizontal percent cover increased progressively across stages. Average tree heights for stages 1–4 were 0 m, 3.6 m, 5.3 m, and 7.2 m, respectively, while average horizontal cover was 0%, 30.2%, 55.2%, and 89.3%.”

Line 276-267: can you combine the first two sentences. Maybe “ June 2024, with increased abundance in 2024 than 2023.”

Line 280: from instead of “form”

Reviewer #2: The manuscript “Effects of woody plant encroachment on abundance of multiple tick species in the U.S. Great Plains” reports the results of comparisons of field collections of ticks from native grassland habitats and habitats classified by three stages of wood plant encroachment. This is an important topic that will be of interest to a broad audience given the course of global change especially in regard to land use and climate. The study presented here is straightforward: ticks were collected in the four habitat classes using two established methods and abundances within species were compared, with two life stages of the most abundant species, Amblyomma americanum, examined separately. Overall, the manuscript is well written and to my judgement, scientifically sound. Most of my comments and suggestions are minor. One thing that is essential is to improve the detail in the Methods regarding the collection methods. This section should include details on the traps that were used, specifics about flagging, and citations of any field techniques or protocols that were used. The conclusions seem to focus on the results related to the two tick species that had higher abundances in habitats with woody plant encroachment, while the results for Amblyomma maculatum seem to be downplayed a bit by the focus on the other two. It is important to describe these results here, and in the Abstract and first paragraph of the Discussion, and to not generalize given the caveat that it was not all tick species that increased in abundance with wood plant encroachment. Overall though, this study is a valuable contribution, well-written, scientifically sound, and interesting.

L40: Mention results for Amblyomma maculatum

L56: Capitalize initial letters of words in names only when proper noun

L68: “effects” should be “affects”

L79: “Eastern redcedar” should be “eastern redcedar”; be consistent with the use of common names and capitalization of initial letters. My suggestion would be that the only words in common names to have an initial capitalized letter should be proper nouns e.g., Bourbon virus, West Nile virus, Bachman’s sparrow, otherwise, they should be, for example, white-tailed deer, white-footed mouse, etc. The alternative would be to capitalize each initial letter e.g., White-tailed Deer. But, it should never be White-tailed deer. Whichever you prefer, be consistent throughout the manuscript. Double check that all other common names are correctly written.

L88: Suggest inserting “those of” between “including” and “birds”

L120: The Methods are well presented, but could use additional detail in some areas. In particular, details about the trapping and flagging need to be included.

L133: If abbreviations are to be used, be consistent with their use. Here, be consistent with use of ERC v. eastern redcedar, and check throughout the manuscript. Also check WPE. Once defined, subsequent mentions of the phrase should be abbreviated.

L136-145: Consider presenting this information as a table

L179: Awkward wording, suggest rewording: “periods when tick activity averages lower than during early summer”

L176-179: Be consistent with date formatting: April 1st – July 15th vs. 16 July – 31 Oct)

L193: Is Ixodes abbreviated here as Ix. for a reason?

L194: Elaborate on the CO2 traps and flagging methods with citations. Were traps commercially purchased or built? Give specifics about flagging. How long were traps left in place and was this consistent across site visits? Were transects linear?

L200: Cite keys that were used.

L209-213: Can this be reworded for clarity? I think I understand what is being communicated, but it took a moment to interpret, especially L210

L221: Could statistical comparisons not be performed on mean tree height and horizontal percent cover between encroachment stage categories?

L228: Were larvae collected?

L240: Spell out Akaike Information Criterion here, put abbreviation in parentheses

L268-270: For consistency, abbreviate names of genera after the first mention of a species name

L280: Change “form” to “from”

L284: Table 1 legend – for clarity, insert “(stage 1)” after “open grasslands”

L308: Would it be meaningful to run this analysis for total Amblyomma americanum? (i.e., Amblyomma americanum adults + nymphs, with trapping method counts combined, as was done for Amblyomma maculatum and Dermacentor variabilis.) Consider including that if it would be meaningful.

L362: Insert “(stage 1)” after “grasslands”

Discussion: The Discussion section is well written, and I appreciate the inclusion of the study’s limitations, especially the potential that trap placement could have biased results. The only weak part of the Discussion is the first paragraph. The language here could be adjusted to make references to trapping methods and stages more clear. It could also be helpful to split this into two paragraphs, with the first focusing summarizing the most important results, (briefly) reiterating the question and the work that was done, and the second explaining the details described in L372-378.

L368: Consider including in this first sentence a statement referencing that Amblyomma maculatum abundances were lower in the Stage 2-4 samples. This result is similarly important as those for the other two species.

L373: Suggest inserting “(overall)” after D. variabilis

L375, 376: For clarity, suggest replacing “collected by trapping” to “collected in CO2-baited traps”

L398: Juniper, or redcedar?

L488: I would include in this initial sentence mention of your results for Amblyomma maculatum as well. It is certainly important that abundance was higher for Amblyomma americanum and Dermacentor variabilis in grasslands with woody plant enchroachment, but it is equally meaningful that other tick species, Amblyomma maculatum, was more abundant in native grasslands. Given that, I would also reword this first sentence a bit – rather than stating that this encroachment facilitates increased abundance of ticks, broadly, I think it would be more accurate to describe this is “some species of ticks”. It is a bit misleading without that caveat.

L493: Would it be worthwhile to rerun your analysis with all three tick species combined? If that were done, you could generalize about ticks a bit more broadly.

L500: Should ERC be WPE?

L509: Was anything known about the fire regime at the study sites? Consider including a note regarding this in the Methods or discussing as a limitation in the Discussion.

L511: “of some species” should be inserted after “increased tick populations”

6. PLOS authors have the option to publish the peer review history of their article (what does this mean? ). If published, this will include your full peer review and any attached files.

**Do you want your identity to be public for this peer review?** For information about this choice, including consent withdrawal, please see our Privacy Policy .

Reviewer #1: **Yes: ** Samniqueka Halsey

Reviewer #2: No

---

## [Author Response · Author response to Decision Letter 1]

7 Jul 2025

Response to Reviewers:

We are grateful for the time that the reviewers have put into making our manuscript better. We have addressed each concern (see bold-face responses below each comment) and hope you find our improved manuscript acceptable for publication

Reviewer #1:

Methods

Statistical Analysis.

Is there a reason for using abundance for the tick drags rather than density of ticks?

AUTHOR RESPONSE: We appreciate this question from the reviewer. We focused on the abundance of ticks collected because our main study questions pertained to a landscape-level spatial scale (i.e., how does a certain area with a specific concentration of ERC impact abundance of ticks that are actively (flagging) and passively (CO2) questing for hosts?). Because all flagging was carried out the same way using different transects for each visit across each study area over the two years, we have a good idea of the abundance of actively-seeking ticks in a given ERC site, which is then comparable with other sites. As we aren’t asking questions related to disease risk (ie. to humans), we didn’t feel that quantifying and analyzing density (ticks/meter) would have addressed our research questions as well as using abundance does.

How are you accounting for the different times of the year that ticks are collected in your analysis?

AUTHOR RESPONSE: Thank you for your question. The primary objective of this study was to assess how tick abundance varies across sites representing four distinct stages of ERC encroachment and across the entire period of seasonal activity for each tick species. To capture these broad patterns, we did not focus on seasonality but instead evaluated abundance across all seasons. Nevertheless, we realize that assessing seasonal variation is an important consideration, and we have added new text in the limitations section of the Discussion (Lines 510-517 in the revised document) describing how we did not consider seasonal variation, and how evaluating hypotheses related to seasonality of ERC effects would be a fruitful area for future research.

Regarding AIC, do you low sample sizes, it might be more appropriate to report AICc, which corrects for bias associated with small sample sizes and is more accurate.

AUTHOR RESPONSE: In response to this reviewer’s later comment related to the relatively small number of models compared, we have decided not to use AIC for the main analysis of the effect of ERC on tick abundance and instead we use ANOVA analyses and p-values linked with pairwise comparisons of group means using the emmeans package. Therefore, this comment is no longer relevant.

Line 247, how are you defining greater support. Is this delta AIC > 2?

AUTHOR RESPONSE: In response to this reviewer’s later comment related to the relatively small number of models compared, we have decided to not use AIC for the main analysis of the effect of ERC on tick abundance. Therefore, this comment is no longer relevant. Instead, we used ANOVA analyses and p-values to identify whether the effect of ERC stage on tick abundance was statistically significant.

Line 249-254: At first read, I wasn’t sure what you mean by stage. I have to reread. Could you use the actual category (grassland, early ERC, moderate ERC and mature ERC. This would make the job of the reader easier.

AUTHOR RESPONSE: We appreciate the concern of the reviewer to make our categories understandable to the general reader. In response to this comment, we have decided to revise (see lines 153-162 in the Methods) to label the 4 categories as: Grassland (control), Early ERC encroachment, moderate ERC encroachment, and mature ERC encroachment (and for shorthand in some tables and figures, we use Grassland, Early ERC, Moderate ERC, and Mature ERC). We also revised to note that these categories correspond to stages 1-4 as defined in another paper that is currently in press and that we believe will be published, and therefore cite-able, before this paper is published.

Also you are considering the grassland as the the reference stage or the control.

AUTHOR RESPONSE: Grassland sites contained no ERC trees at the time of sampling; however, as we cannot guarantee that ERC encroachment had not already begun in grassland sites (in the form of ERC seeds deposited in the seedbank or on the ground), we think it is safer to refer to these as control sites and not reference sites (since reference sites generally imply pristine or idealized conditions, which in our case would mean grasslands free of any influence of ERC or its seeds). We have revised in the study design section to clarify that we considered open grassland sites to be our control treatment.

I’m not sure what are you running separate models with each of the other stages of ERC encroachment as the reference for the intercept. To perform post comparisons, it would be more appropriate to use emmeans package as this would adjust the p-values for multiple comparisions?

AUTHOR RESPONSE: Thank you for bringing this to our attention. As noted in other comments, we have revised to use an ANOVA and p-value testing approach for all of our analyses, now also linked with use of the emmeans package to conduct a single set of pairwise comparisons for each response variable. This allowed us to run a single model for each response variable and to account for the potential biasing effect of multiple statistical comparisons.

Results

Table 2: I’m sure what you are using AIC when there are only two models to compare, one of which is the null. Usually you would have multiple predictors that you compare. In doing this, it would make more sense to just report the p-values.

AUTHOR RESPONSE: We have addressed this comment by changing to use ANOVA analyses and p-values to assess the significance of the effect of ERC stage for each tick abundance response variable. Also as noted above, we now make comparisons between all group means for each analysis by using the emmeans package and post hoc pairwise comparisons. As a result of these changes, Table 2 is no longer needed

Figures: for the X axis, instage of putting Stage 1,2,3 can you use “Open grassland”, “early ERC”, etc.

AUTHOR RESPONSE: Although we have used these stage identifiers in previously published papers for the same study system, we appreciate the reviewers concern, and we have decided to revise to label the 4 categories as: Grassland (control), Early ERC encroachment, moderate ERC encroachment, and mature ERC encroachment (and for shorthand in some tables and figures, we use Grassland, Early ERC, Moderate ERC, and Mature ERC). We also revised to note that these categories correspond to stages 1-4 as defined in another paper that is currently in press and that we believe will be published before this paper is published.

When describing the results, can you put it in context, instead of just stating the beta values, transform this into a percentage change. For instance in lines 318-319, based on the beta values, and if you are using a log link, the beta value can be convert indicates that "A. americanum was about 5 times as abundant in stage 2 as in stage 1," or ~400% more abundant, relative to stage 1."

AUTHOR RESPONSE: We agree that statements about the magnitudes of differences between group means can add valuable context about the ecological significance of statistically significant differences. To address this comment, we have added such statements (e.g., at Lines 315-317 in the Results) about quantitative differences between tick abundances in different stages (particularly between grasslands and early ERC encroachment sites, which was the most frequent statistically significant difference that we documented).

I’m not sure how you got the letter values for the figures.

AUTHOR RESPONSE: As we had described in the captions for the bar graphs, letters shared among groups represent groups that are not significantly different from each other and letters that were different between groups indicate groups that are significantly different from each other. This is a standard convention across many scientific fields, as evidenced by the large number of online references and tutorials about this topic (here: https://www.google.com/search?client=firefox-b-1-d&q=letters+on+bar+graphs+to+indicate+significant+differences). If the journal’s editorial team has guidance or opinions on how better to present these graphs, we would of course defer to their recommendations.

What is a coefficient analysis? See above comments about using emmeans.

AUTHOR RESPONSE: Thank you for catching this lack of clarity. We meant that we had used coefficients from model parameters for the ERC stage fixed effect to estimate stage group means. We have revised this section (now at Lines 264-271 in methods) to clarify and to update this description to match the new emmeans analysis which generates estimates of group means and statistical significance of differences between them.

Throughout the manuscript, you have a tendency to have the same information (AIC values, beta values) in the text that is in a table. Choose one,

AUTHOR RESPONSE: Due to the change in our analysis approach to use ANOVA linked with pairwise comparisons, the tables with repeated information have been deleted. All information is now presented in either text or figure format, but not both.

Discussion.

The discussion is well written given the current results, I won’t provide speciific comments as it is unknown if it will change when the method/results change.

AUTHOR RESPONSE: We appreciate the assessment of the reviewer. The changes made regarding the statistical analysis did not produce any major changes to the storyline. For all analyses, we have still found a significant difference in abundance between grassland (control) sites and the earliest stage of ERC encroachment, although some of the previously shown differences between the 3 stages with ERC trees are no longer significant (i.e., all 3 stages of ERC encroachment have greater abundance than grasslands, but the 3 stages of encroachment are not significantly different from each other). Thus, the major conclusion that even the earliest stage of ERC encroachment increases tick abundance remains unchanged. That said, we have made some minor revisions in the discussion to align with our updated analyses and results.

Minor Comments

Line 168L misspelling “centeral”

AUTHOR RESPONSE: Spelling corrected as suggested.

Lines 179: be consistent with the date format, earlier, you put the month day and on this line you put day month.

Be consistent with numbers, when using numbers less than 10, spell out the entire word.

AUTHOR RESPONSE: Date format has been changed to be more consistent. We have also revised to make sure we refer to numbers correctly throughout the paper, although we note that this is more of a copy editing issue and will defer to the journal editorial staff for guidance on such issues.

Line 258-263: what there an actually analysis for this? Also, the tect could be a bit more concise “Measurements of ERC trees confirmed that our sampling captured distinct stages of encroachment: both tree height and horizontal percent cover increased progressively across stages. Average tree heights for stages 1–4 were 0 m, 3.6 m, 5.3 m, and 7.2 m, respectively, while average horizontal cover was 0%, 30.2%, 55.2%, and 89.3%.”

AUTHOR RESPONSE: We appreciate this comment and your suggested rewording. To clarify, there was no formal analysis run on measurements of ERC tree height and horizontal percent cover. Our goal was not to make this analysis a centerpiece of the study, but rather, to make it clear that our sites in different stages had some real separation in terms of the tree characteristics that likely influence tick abundance. To better introduce that this was a descriptive analysis, we have revised the Methods section that describes these two metrics to note that this was a descriptive comparison (Lines 220-224). In the results section referenced by the reviewer, and in addition to incorporating the above text revision suggestions, we have also revised the lead-in text to remind the reader that these are descriptive comparisons of stage means for ERC tree height and horizontal cover.

Line 276-267: can you combine the first two sentences. Maybe “ June 2024, with increased abundance in 2024 than 2023.”

AUTHOR RESPONSE: Text was revised as suggested.

Line 280: from instead of “form”

AUTHOR RESPONSE: Spelling corrected as suggested.

Reviewer #2:

L40: Mention results for Amblyomma maculatum

AUTHOR RESPONSE: Thank you for this comment. We revised to add results for Amblyomma maculatum to the abstract.

L56: Capitalize initial letters of words in names only when proper noun

AUTHOR RESPONSE: Corrected as suggested.

L68: “effects” should be “affects”

AUTHOR RESPONSE: Corrected as suggested.

L79: “Eastern redcedar” should be “eastern redcedar”; be consistent with the use of common names and capitalization of initial letters. My suggestion would be that the only words in common names to have an initial capitalized letter should be proper nouns e.g., Bourbon virus, West Nile virus, Bachman’s sparrow, otherwise, they should be, for example, white-tailed deer, white-footed mouse, etc. The alternative would be to capitalize each initial letter e.g., White-tailed Deer. But, it should never be White-tailed deer. Whichever you prefer, be consistent throughout the manuscript. Double check that all other common names are correctly written.

AUTHOR RESPONSE: Thank you for catching this inconsistency with the use of common names and capitalization. We went through and corrected all common names to have consistent capitalization. Please also note that we will of course defer to the journal editorial staff on such formatting issues, should we reach the copy-editing stage for this paper,

L88: Suggest inserting “those of” between “including” and “birds”

AUTHOR RESPONSE: Text has been revised based on your comment.

L120: The Methods are well presented, but could use additional detail in some areas. In particular, details about the trapping and flagging need to be included.

AUTHOR RESPONSE: In response to the reviewer’s suggestion, we have added more details in regard to CO2 trapping.

L133: If abbreviations are to be used, be consistent with their use. Here, be consistent with use of ERC v. eastern redcedar, and check throughout the manuscript. Also check WPE. Once defined, subsequent mentions of the phrase should be abbreviated.

AUTHOR RESPONSE: We have gone through and changed all subsequent eastern redcedar to ERC and woody plant encroachment to WPE.

L136-145: Consider presenting this information as a table

AUTHOR RESPONSE: We have revised to add a table as suggested, including the first column showing listing the two broad land cover types we sought to identify (grassland and ERC tree cover) and the second column listing the specific cover categories included in the search for each broad land cover type.

L179: Awkward wording, suggest rewording: “periods when tick activity averages lower than during early summer”

AUTHOR RESPONSE: Reworded for clarity as suggested.

L176-179: Be consistent with date formatting: April 1st – July 15th vs. 16 July – 31 Oct)

AUTHOR RESPONSE: Text has been revised as suggested.

L193: Is Ixodes abbreviated here as Ix. for a reason?

AUTHOR RESPONSE: Thank you for catching this inconsistency. All abbreviations of Ixodes have been revised to be “I.”.

L194: Elaborate on the CO2 traps and flagging methods with citations. Were traps commercially purchased or built? Give specifics about flagging. How long were traps left in place and was this consistent across site visits? Were transects linear?

AUTHOR RESPONSE: We have added the requested details, including noting that transects for flagging were linear and that our flagging approach followed widely used CDC guidelines, as well as additional details about the construction and operation of CO2 traps.

L200: Cite keys that were used.

AUTHOR RESPONSE: Citations were added as suggested.

L209-213: Can this be reworded for clarity? I think I understand what is being communicated, but it took a

---

## [Decision Letter · Decision Letter 1]

5 Sep 2025

Effects of woody plant encroachment on abundance of multiple tick species in the U.S. Great Plains

PONE-D-25-16397R1

Dear Dr. Propst,

We’re pleased to inform you that your manuscript has been judged scientifically suitable for publication and will be formally accepted for publication once it meets all outstanding technical requirements.

Kind regards,

Xiao Guo, Ph.D.

Academic Editor

PLOS ONE

Additional Editor Comments (optional):

The authors have adeptly and thoroughly addressed the reviewers' concerns, and the manuscript's overall quality has been substantially enhanced. The paper is well-structured, its arguments are lucid and coherent, and I therefore recommend that it be accepted for publication.

Reviewers' comments:

Reviewer's Responses to Questions

**Comments to the Author**

1. If the authors have adequately addressed your comments raised in a previous round of review and you feel that this manuscript is now acceptable for publication, you may indicate that here to bypass the “Comments to the Author” section, enter your conflict of interest statement in the “Confidential to Editor” section, and submit your "Accept" recommendation.

Reviewer #1: All comments have been addressed

Reviewer #2: All comments have been addressed

2. Is the manuscript technically sound, and do the data support the conclusions?

Reviewer #1: Yes

Reviewer #2: Yes

3. Has the statistical analysis been performed appropriately and rigorously? 

Reviewer #1: Yes

Reviewer #2: Yes

4. Have the authors made all data underlying the findings in their manuscript fully available?

Reviewer #1: Yes

Reviewer #2: Yes

5. Is the manuscript presented in an intelligible fashion and written in standard English?

Reviewer #1: Yes

Reviewer #2: Yes

6. Review Comments to the Author

Reviewer #1: Thank you for addressing the review comments. I have no further concerns at this time and believe the manuscript is suitable for publication.

Reviewer #2: (No Response)

7. PLOS authors have the option to publish the peer review history of their article (what does this mean? ). If published, this will include your full peer review and any attached files.

**Do you want your identity to be public for this peer review?** For information about this choice, including consent withdrawal, please see our Privacy Policy .

Reviewer #1: **Yes: ** Samniqueka Halsey

Reviewer #2: No

---

## [Editor Report · Acceptance letter]

PONE-D-25-16397R1

PLOS ONE

Dear Dr. Propst,

I'm pleased to inform you that your manuscript has been deemed suitable for publication in PLOS ONE. Congratulations! Your manuscript is now being handed over to our production team.

Kind regards,

on behalf of

Dr. Xiao Guo

Academic Editor

PLOS ONE